# COVID-19 Mortality and Related Comorbidities in Hospitalized Patients in Bulgaria

**DOI:** 10.3390/healthcare10081535

**Published:** 2022-08-14

**Authors:** Rositsa Dimova, Rumyana Stoyanova, Vesela Blagoeva, Momchil Mavrov, Mladen Doykov

**Affiliations:** 1Department of Health Management and Health Economics, Medical University of Plovdiv, 4002 Plovdiv, Bulgaria; 2First Department of Internal Diseases, Section of Pneumology and Phthisiatrics, Medical University of Plovdiv, 4002 Plovdiv, Bulgaria; 3Department of Healthcare Management, Section of Medical Ethics and Low, Medical University of Plovdiv, 4002 Plovdiv, Bulgaria; 4Department of Urology and General Medicine, Medical University of Plovdiv, 4002 Plovdiv, Bulgaria

**Keywords:** COVID-19, pandemic, multiple chronic conditions, comorbidity, incidence, case fatality rate

## Abstract

Until now, the COVID-19 pandemic has resulted in at least 27 million cases and over 900,000 deaths worldwide. Bulgaria is one of the countries that is the most severely affected by the COVID-19 pandemic, and the mortality rate is among the highest registered in the world. The aim of this study is to investigate and analyze mortality rates due to the fact of COVID-19 in addition to the most common related underlying medical conditions in those hospitalized to outline the factors that have an impact on the mortality rate due to the fact of COVID-19. A descriptive cross-sectional research design with a retrospective analysis was used to collect data from a total of 128,269 hospitalized patients during the period from April 2020 to November 2021. During the study period, the number of hospital admissions due to the fact of COVID-19 was 5200. The patients’ mean age was 67.34 (SD ± 19.65), and 51.7% (2689) of the patients were men. Only 10% of out of a total of 5200 patients did not test positive for COVID-19 upon admission based on the antigen or PCR test. Out of all patients, 41.5% had no underlying medical conditions upon presentation, and the remaining 58.5% had diagnosed comorbidities, varying from one to five. One-third (1470) had a lethal outcome, and the remaining 71.7% recovered from the infection and were discharged from the hospital. Based on the analysis of our results, there is definite evidence that the CFR and susceptibility to symptomatic COVID-19 were higher in the elderly, men, and patients with more comorbidities, especially chronic cardiovascular, metabolic, and respiratory disorders, as well as in those admitted to hospital within 6 h after an emergency ward visit and who had a shorter mean hospital stay.

## 1. Introduction

Until now, the COVID-19 pandemic has resulted in at least 27 million cases and over 900,000 deaths worldwide [1]. There are two methods used to assess the proportion of infected individuals with fatal outcomes: the infection fatality ratio (IFR), which estimates the proportion of deaths among all infected individuals, and the case fatality rate (CFR), which estimates the proportion of deaths among identified confirmed cases [2]. The official data unveil that the CFR of SARS-CoV among the general population is 1.0% compared to the CFR of seasonal flu in the US, which is approximately 0.1% to 0.2% [3,4]. A number of recent data confirm the presence of an excess in mortality, i.e., deaths from all causes compared to the average rate over previous years or the percentage difference between the reported number of weekly or monthly deaths in 2020–2021 and the average number of deaths in the same period over the years 2015–2019 [5,6]. According to the World Health Organization’s (WHO) statistics regarding the COVID-19 pandemic, a significant upward trend in the overall weekly or monthly mortality rate across the world in 2020 was registered, compared to the preceding three-year period, i.e., an excessive mortality rate of at least 3,000,000 was documented [7]. A preliminary analysis of the data show that worldwide incidence rates of COVID-19 are almost equal among men and women—48% versus 51%, respectively, with some variations across age groups [8]. However, whereas males and females have the same prevalence of COVID-19, men with COVID-19 are at a higher risk for worse outcomes and death, regardless of age [9]. The global data indicate higher COVID-19 case fatality rates among men than women [10]. Most countries with available data indicate a male-to-female case fatality ratio higher than 1.0, ranging up to 3.5 in some cases [10]. Based on the Centers for Disease Control and Prevention statistics in 2021, the incidence rate of COVID-19 (85%) was the highest in the 18–49 age group [11]. However, the percentage of deaths increased with age, and approximately 75% of deaths were in those aged 65 years and older [12]. Compared to those 18 to 29 years old, the rate of death was four times higher in those 30 to 39 years old, 25 times higher in those 50–64 years old, and 340 times higher in those who were 85 years of age or older [5]. Another similar study revealed that the mortality rate was the highest in the age group over 65 years, which accounted for 52% of all lethal COVID-19 cases [11]. It was also found that older age and a high number of comorbidities were associated with higher severity and mortality in patients with COVID-19 [9].

Unfortunately, Bulgaria is among the countries that is the most severely affected by the COVID-19 pandemic, and the mortality rates are among the highest registered in the world. An excess death rate per 100,000 population has been documented. According to a Eurostat report, which compared additional deaths with pre-pandemic levels, Bulgaria has one of the highest excess mortality rates, peaking in November 2020 (94.0%). The annual excess mortality for the EU as a whole, in 2020, was higher by 11.9% than the 2016–2019 average; in 2021, it was higher by 14.3%. There was a wide range of variation during the same year period among the Member States and in the same Member States on an annual basis. In 2021, Sweden and Belgium had an annual excess mortality rate below 5%, in contrast to Bulgaria, where the reported annual excess mortality rate was close to 40%. The two countries that registered the most substantial improvements in the second year of the pandemic were Belgium (from an annual excess mortality rate of 16.2% in 2020 to 3.3% in 2021) and Spain (from 18% in 2020 to 7.7% in 2021). Compared to them, unfortunately, in Bulgaria, the annual excess mortality rate increased in 2021 compared to 2020, from 14.6% to 37.7% [13]. Statistical reports from the Johns Hopkins Center for Research on the coronavirus show a mortality rate of 369.19 per 100,000 of the Bulgarian population, which ranks among the highest worldwide [14]. These data provided us with the grounds to investigate, in detail, the documented excessive mortality in the country. In Bulgaria, so far no similar studies have been published, which gave us further grounds to plan and conduct this study.

The aim was to investigate and analyze the mortality rate due to the fact of COVID-19 and the most common related underlying medical conditions in hospitalized patients at St. George University Hospital in Plovdiv as well as to outline the factors that have an impact on the mortality rate due to the fact of COVID-19.

## 2. Materials and Methods

A descriptive cross-sectional research design with a retrospective analysis was used to collect data from a total of 128,269 patients, who were hospitalized for various medical diagnoses at St. George University Hospital in Plovdiv. The WHO definitions for confirmed, suspected, or probable cases of COVID-19 were used to recruit patients in the study [15]. Clinically suspicious cases had an antigen rapid detection test (RDT) performed in the emergency ward. Positive, mild cases were referred to their general practitioners for treatment and monitoring. Moderate, severe, and critical cases of COVID-19 were admitted to the specialized wards and, additionally, had a PCR performed. A total of 5200 patients were enrolled, and only 10% of them had epidemiological, clinical, and imaging studies suspicious for COVID-19 but who did not produce a positive antigen or PCR test. A review of the patients’ medical records was performed covering the period from April 2020 to November 2021. The documentary method was employed to accumulate primary and secondary information (document analysis from the WHO and the National Statistics Agency databases). According to the WHO requirements, the mortality rate was calculated based on the immediate cause of death—infection with COVID-19, either U07.1 COVID-19 or U07.2 COVID-19 as registered in the International Classification of Diseases (ICD-11). The inclusion criteria were the WHO case definitions of a confirmed case of SARS-CoV-2: either (1) a patient with a positive nucleic acid amplification test (NAAT) test, regardless of the clinical criteria or epidemiological criteria, or (2) a patient meeting the clinical criteria and/or epidemiological criteria (i.e., suspected case) with a positive professional-use or self-test SARS-CoV-2 antigen RDT. There were no exclusion criteria [15].

All requirements regarding confidentiality of medical and personal data were strictly followed when collecting and analyzing the data according to the General Data Protection Regulations (GDPR) (EU) 2016/679 issued by the European Parliament.

Other methods applied were descriptive statistics (i.e., number, percentage, mean, and standard deviation), nonparametric tests (i.e., Fisher’s exact test, Pearson chi-square test, and the Mann–Whitney test), correlation coefficients (i.e., Spearman correlation), and logistic regression. The enrollment of patients for the current study is presented in Figure 1.

## 3. Results

During the study period, hospital admissions due to the fact of COVID-19 were 5200. Of them, 5161 (99.2%) had a history, symptoms, and imaging tests suggestive for COVID-19 upon their visit to the emergency ward. Thirty-nine patients (0.8%) presented with other symptoms and were admitted for medical diagnoses other than COVID-19 but at a later stage produced a positive antigen or PCR test. The mean age of the patients was 67.34 (SD ± 19.65), with a lower age limit of under one year of age and an upper limit of 99 years of age. Of them, 51.7% (2689) were men. Emergent hospitalization is defined as the time elapsed from an emergency ward visit to hospitalization due to the fact of infection with COVID-19.

It should be noted that out of 5200 admitted patients, 90% were confirmed with a PCR test. Only 10% who presented with symptoms and had a laboratory constellation and imaging studies typical for COVID-19-related pneumonia did not test positive for COVID-19 based on the PCR test. Out of 5200 patients, 41.5% had no underlying medical conditions upon presentation (Table 1). The remaining had from 1 to 5 diagnosed comorbidities. The most common accompanying medical conditions were cardiovascular disorders, followed by endocrine and respiratory (Table 1).

Of all hospitalized patients with COVID-19, approximately one-third (28.3%) (n = 1470) had a lethal outcome, the remaining 71.7% recovered from the infection and were discharged from the hospital. Follow up on these patients was performed by their general practitioners.

The average number of comorbidities in our contingent of patients was 1.32. A statistically significant difference was established between the number of comorbidities and the disease outcome. In discharged and recovered patients, this indicator was twice as low compared to that of patients with a lethal outcome (*p* = 0.000) (Table 2).

Fisher’s exact test documented that the lethality rate was higher in men compared to women (*p* = 0.000).

The Mann-Whitney test was used to document that a patient’s age had a negative impact on the disease outcome. There was a statistically significant difference between the age of patients and lethal outcome compared to discharged survivors (U = 1,542,330; *p* = 0.000). Discharged patients presented at a significantly younger mean age of 57.3 ± 20.50 years. The mean age of patients with a lethal outcome was significantly higher at 71.3 ± 12.26 (Table 2).

The maximum duration of hospital stay was 113 days. Nonparametric analysis showed a statistically significant difference between the mean hospital stay of survivors compared to those with a lethal outcome. Patients with a lethal outcome had a shorter mean hospital stay compared to discharged patients (U = 1,894,653; *p* = 0.000).

The lethality rate in patients with a diagnosed COVID-19 infection was 28.4%. Patients admitted for other medical conditions but who at a later stage were diagnosed with COVID-19 had a lethality rate of 5.1% (*p* = 0.000).

The time elapsed from a patient’s visit to the emergency ward to hospital admission was also found to impact mortality. It was higher in patients admitted within 6 h after the emergency visit at −30.3% compared to those admitted after 6 h at −25.3% (χ^2^ = 10.010; *p* = 0.007).

There was a correlation between the number of comorbidities and the disease out-come in COVID-19. The nonparametric Pearson chi-square coefficient showed that multiple underlying medical conditions increased the likelihood of a lethal outcome (χ^2^ = 497.582; *p* = 0.000). The same relationship was confirmed based on Spearman correlation analysis, which documented a moderate to strong correlation (rs = 306; *p* = 0.000).

It was also established that the nature of comorbidities influenced the mortality rate of COVID-19. Rheumatological disorders were less frequently related with mortality in patients with COVID-19 (χ^2^ = 43.226; *p* = 0.000). Of all fatalities, the percentage was the lowest in patients with concomitant rheumatological conditions at −0.5%, whereas it was the highest in patients with underlying cardiovascular disorders at 70.1%, followed by endocrinological and other metabolic disorders at −15.4%.

The results of the significance test in the regression model was χ^2^ = 872.699, df = 4, *p* = 0.000, whereas the established result of the Hosmer–Lemeshow test was 11.941 (*p* = 0.154; *p* > 0.05), indicating optimal regression models. This model explained 22.2% of the statistical dispersion and adequately classified 71.7% of the dependent variable observations, i.e., the mortality rate. The four independent variables—sex, age, number of comorbidities, and emergent medical admission—were found to be statistically significant based on the Wald criterion (*p* < 0.001), (Table 3).

Each additional year of age increased the likelihood of a lethal outcome by 5% (Waldχ^2^ = 328.86; *p* = 0.00; OR = 1.050). Women had a lower risk of a lethal outcome compared to men (Waldχ^2^ = 29.702; *p* = 0.00; OR = 0.691). Patients admitted with an initial diagnosis of COVID-19 had a seven times higher likelihood of a fatal outcome compared to those admitted for other diagnoses and who were diagnosed with a COVID-19 infection at a later stage (Waldχ^2^ = 6.974; *p* = 0.08; OR = 7.213).

## 4. Discussion

Based on analysis of our results, there is definite evidence that the CFR and susceptibility to symptomatic COVID-19 was higher in the elderly, men, and patients with more comorbidities, especially chronic cardiovascular, metabolic, and respiratory disorders. The exact nature of the greater severity of COVID-19 in these patients is not yet clearly understood. There are different explanations, from aging of immunity and its inability to control the virus in the elderly to the more severe general inflammation in this population. On the other hand, it is possible that women are able to build a better immune response than men or that men tend to engage in more risky behaviors and seek medical aid at a later stage. Regarding comorbidities, cardiovascular disorders and diabetes seemed to increase the risk more significantly due to the presence of related underlying vasculitis, which seemed to increase ischemic damage, or there was direct viral infection in the myocardium. Further research is needed in this area.

Based on sources from the literature, COVID-19 fatality rates in the general population vary significantly depending on the country, from 0.06% in Qatar to 16.25% in Belgium as of 26 May 2020 [3,16]. In another study, the CFR in hospitalized patients was 13.0%, whereas our study showed a CFR of 28.4% [3]. Death and the severity of COVID-19 were associated with age, comorbidities, and the availability of early access to medical care. This varies across the world [17]; however, there is a consistent and clear pattern of an age-based exponential increase in fatality rates, regardless of the geographic region [18]. Among COVID-19 patients, elderly patients had a higher mortality rate due to the fact of a high CFR and symptomatic infection rate. Approximately 80% to 90% of deaths occurred in patients aged >70 years. These data are consistent with an observed tendency for this worldwide. Most fatalities occurred in the age group over 80 years. Different explanations are available in the literature; however, the most likely of them are in relation to the immune-senescence factor [18,19,20]. With age, the production of naive T and B cells decreases, and the function of innate immune cells is impaired; hence, cells involved in innate immunity are not activated efficiently during an infection, and progression to an adaptive immune response does not occur in a coordinated manner [18,19,20]. These changes reduce the effectiveness of viral clearance and increase the likelihood of triggering a dysregulated immune response in which cytokines are released extensively by activated immune cells, resulting in a cytokine storm. Another well-recognized feature of aging immunity is chronic subclinical systemic inflammation, also known as inflammaging. Inflammation is a key pathogenic mechanism in COVID-19; hence, inflammaging has been estimated to contribute to the poorer outcome in elderly patients with COVID-19 [21]. In addition to aging immunity, there are several other factors related to aging that could be reasons for the higher mortality and morbidity in the elderly. The average number of comorbid conditions steadily increases with age [18]. Our results similarly documented that comorbidity was a risk factor for mortality—it steadily increased as the number of underlying medical conditions increased [18]. Similar to our results, other authors have discovered that older age and a high number of comorbidities are associated with a higher severity and mortality in patients with both COVID-19 and SARS [9]. Other important features of COVID-related mortality are that men are more severely affected in terms of COVID-19-related fatalities [14], more men than women have died of COVID-19 in 41 out of 47 countries [22], and the overall COVID-19 case-fatality ratio is approximately 2.4 times higher among men than among women [23,24]. However, there exist data that show that COVID-19 the case-fatality rate is higher in women than men in a few countries, for instance, India, which is one of the most affected countries [14]. In the largest survey of 72,314 suspected or confirmed cases of COVID-19 in China (men, 63.8% of cases; women, 36.2% of cases), the case-fatality ratio was higher among men (2.8%) than among women (1.7%) [21]. Another study from China on critically ill patients showed that men with comorbidities, such as hypertension, cardiovascular disease, chronic kidney disease, and diabetes, had the highest mortality [25], and US data show similar patterns [20,23,24]. Based on our results, COVID-19 infected men and women equally at 48.7% and 51.3%, respectively. However, the mortality rate was significantly higher in men at −57.6%. At present, there is no clear explanation for this phenomenon. Biological and psychosocial factors have been implicated. Some researchers attribute it to the X chromosome, which contains a high density of immune-related genes; therefore, women generally mount stronger innate and adaptive immune responses than men [26]. However, psychosocial factors also seem to play a role: men usually engage in more risky behaviors, seem to downplay the risks, and usually seek medical help later compared to women [27]. We believe that in our country, similar to data on the US, these factors also seemed to be responsible for the higher rate of mortality in men.

Another important finding in our study is the strong correlation between comorbidities and COVID-19 mortality. As seen from the results, the average number of underlying medical conditions in our sample is 1.32. A statistically significant difference was established between the number of concomitant disorders and the disease outcome. It was also established that the nature of the underlying medical conditions influenced the mortality rate of COVID-19. Rheumatological disorders were less frequently related with mortality in patients with COVID-19—0.5%, whereas it was the highest in cardio-vascular disorders—70.1%, followed by endocrinology and other metabolic disorders—15.4%.

Meta-analyses from China and Spain also suggested that age and comorbidities were highly related in COVID-19 patients [27,28]. Of these deceased patients, 64.9% had at least one underlying disorder (e.g., hypertension, diabetes, cardiovascular disease, or chronic obstructive pulmonary disease) [29]. A meta-analysis in the US concluded that comorbidities as hypertension, diabetes, cardiovascular diseases, and respiratory system diseases are a risk factor for severe outcomes compared to non-severe outcomes. Across all patients, significant comorbidities included hypertension (43.6%), diabetes (23.8%), and coronary heart disease (CHD)/cardiovascular disease (12.4%). Similarly, in our study, cardiovascular disorders ranked first, followed by diabetes and respiratory disorders. On the other hand, the US study did not state malignant and rheumatology disorders as risk comorbidities in COVID-19 patients [30]. Regardless of the specified comorbidities, all studies found that cardiovascular disorders and diabetes were related with higher mortality in COVID-19 patients. They accounted for more than half of the lethal outcomes. The US meta-analysis outlined a probable correlation between cardiovascular disease and COVID-19. This is attributable to infection-related ischemia, resulting in myocardial injury and/or a viral-induced inflammatory storm causing shock. However, there is increasing evidence that COVID-19 might cause a direct viral infection of the myocardium [31,32]. Finally, the mean duration of hospital stay in our study was found to be 10 days. In other studies, it ranged from 14 in China to 7 days in most countries [32]. A difference was established with patients who were discharged alive having a longer stay than those who died. This tendency was also observed in our study. The likely explanation for this is the more severe medical condition upon admission of diseased patients.

## 5. Conclusions

The study confirms that the outcome of an infection with COVID-19 depends on the age, sex, and number of comorbidities as well as on the type of hospitalization: emergent or planned. In our opinion, our findings are valuable and timely for decision makers to develop strategies for reducing morbidity and mortality due the fact of COVID-19. Our study revealed that mortality was higher in men compared to women, in older age groups, when the patients were hospitalized from the emergency ward up to the 6th hour, and when the number of comorbidities was high, especially in cardiovascular disorders and diabetes.

From a clinical perspective, our study will help clinicians in decision making (at present, monoclonal antibodies are available for high-risk groups, and it should be administered as soon as possible) and will guide them when choosing the proper treatment. We believe that the age, sex, and the number and character of comorbidities are important predictors of the infection’s course and outcome and will aid in the development of reliable severity scores similar to those used for pneumonia. Moreover, this study illustrated that access to high-quality medical treatment and the organization of healthcare also play important roles in disease outcome.

## Figures and Tables

**Figure 1 healthcare-10-01535-f001:**
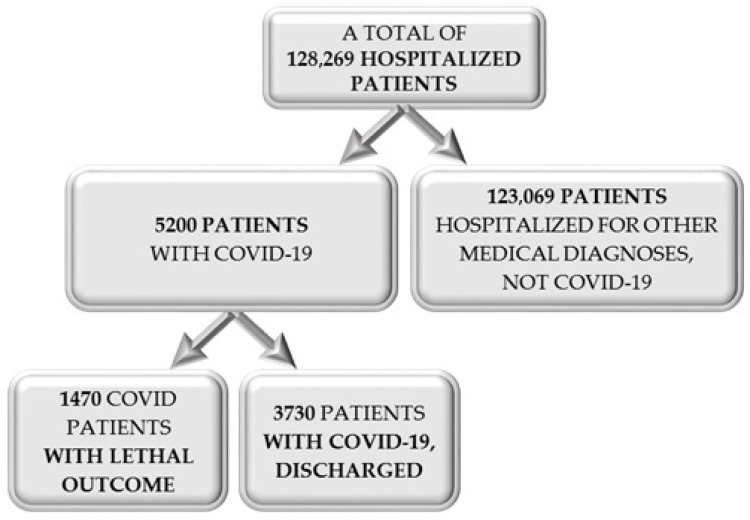
Flow chart of the participants’ enrollment.

**Table 1 healthcare-10-01535-t001:** Medical characteristics of the patients upon their admission to the hospital.

Variable		n	%
** *Emergent or elective hospital admissions* **	Up to 6 h	1015	19.5
	Up to 12 h	674	13
	Up to 24 h	783	15.1
	After 24 h	2689	51.7
	Elective admission	39	0.8
	** *Total* **	5200	100
** *Number of comorbidities* **	No underlying conditions	2160	41.5
	One	1275	24.5
	Two	768	14.8
	Three	458	8.8
	Four	249	4.8
	More than five	290	5.6
	** *Total* **	5200	100
** *Character of comorbidities* **	Cardiovascular disorders	1583	67.7
	Endocrine and other metabolic disorders	275	11.8
	Lung disorders	94	4.0
	malignancy	82	3.5
	Gastrointestinal disorders	71	3.0
	Genitourinary disorders	55	2.4
	Neurological	38	1.6
	Rheumatological	20	0.9
	Other	121	5.2
	** *Total* **	2339	100

**Table 2 healthcare-10-01535-t002:** Patients’ characteristics and disease outcome.

Patients’ Characteristics and Disease Outcome	Total		Discharged		Lethal Outcome		*p*
** *Sex* **	**N**	**(%)**	**n**	**(%)**	**n**	**(%)**	0.000
** *Male* **	2689	(100.0)	1860	(69.2)	829	(30.8)	
** *Female* **	2511	(100.0)	1870	(74.5)	641	(25.5)	
** *The mean age of patients* **	** *Mean ± SD* **		** *Mean ± SD* **		** *Mean ± SD* **		0.000
	67.34 ± 19.65		57.3 ± 20.50		71.3 ± 12.26		
** *The mean number of* ** ** *comorbidities* **	** *Mean ± SD* **		** *Mean ± SD* **		** *Mean ± SD* **		0.000
	1.32 ± 1.594		1.03 ± 1.419		2.05 ± 1.770		
** *The mean duration of hospital stay* **	** *Mean ± SD* **		** *Mean ± SD* **		** *Mean ± SD* **		0.000
	10.2 ± 7.54		11.08 ± 7.30		8.13 ± 7.70		

**Table 3 healthcare-10-01535-t003:** Regression analysis of the factors that influenced disease outcome.

	B	SE	Wald	df	*p*	OR	95% CI for OR	
							Lower	Upper
** *Sex* **	−0.369	0.068	29.702	1	0.000	0.691	0.605	0.789
** *Age* **	0.048	0.003	328.862	1	0.000	1.050	1.044	1.055
** *Emergency* ** ** *admission* **	1.976	0.748	6.974	1	0.008	7.213	1.664	31.262
** *Number of comorbidities* **	0.246	0.021	142.221	1	0.000	1.279	1.228	1.332
** *Constant* **	−6.272	0.768	66.727	1	0.000	0.002		

## Data Availability

Data are available upon request with restrictions, e.g., privacy or ethical considerations. The data presented in this study are available upon request from the corresponding author.

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
