# Peer review of "COVID-19 Mortality and Related Comorbidities in Hospitalized Patients in Bulgaria"

_healthcare, 2022, doi:10.3390/healthcare10081535_

Round 1

Reviewer 1 Report

I think it's a good written paper. However, authors might want to revise:

1) The uses of "death" (line 68), "lethal" (line 124), and "mortality"(line 82)... rate. Authors used three different words but I believe they had the same meaning. As a reader, it is quite confusing. Maybe, choose only one word and use it consistently throughout the paper.

2) Line 85 stated that data were collected from 128 269 patients, but in line 106, data were collected from 5200 patients. Please check for consistency.

3) Literature Review section should be added because there is no explanation about why authors want to do this research, what similar researches have been done before, how authors' research is different from others, and much other information to convince academic research perspectives. It is not appropriate to suddenly know that there were many research findings supporting authors' research findings (in the discussion section). Hence, how authors' research is different from previous research and can contribute to the body of knowledge?

4) The Research Impact section should be added because it is very important to highlight how authors' research is significant and needed. I did not see any explanation how authors' findings can be useful for others either for researchers or practitioners.

Author Response

Dear Editors,

Thank you for the detailed reviews and the suggested amendments to out article.

We are grateful to reviewer 1 for the favorable review. Regarding the suggested revision, our opinion is as follows:

Answer to Question 1: Definitely the word “death “is not appropriate. If it is present in the manuscript, this should be corrected. However, we believe that lethality and mortality bear a different meaning. Mortality refers to the number of deaths in a particular area or from a particular cause: lethality is more related to the capacity to cause death or a serious damage, therefore they are not synonyms. It seems that it is appropriate to keep mortality when referring to the number of deaths due to COVID - 19 on a large scale and keep lethality and lethal outcome when referring to the clinical outcomes of the infection to illustrate the potential of the virus to cause death in the clinical cases.

Answer to Question 2: The number 128 269 refers to all hospital admissions at St. George Hospital for all causes. 5200 is the number of hospital admissions due to COVID -19. Both figures refer for the stated period of time. If this is not clear in the text, we shall clarify it.

Answer to Question 3: We agree with this remark. In Bulgaria, so far no similar researches have been published. The reason why we planned and conducted the survey is the fact that the COVID -19 related mortality and morbidity in Bulgaria was extremely high. However, results from similar studies show a high variability of mortality rates – from 0.06 to 16.25   based on different sources [3,14]. Is this due to methodology reasons only or the structure and the general morbidity in the population influences this to some extent, or healthcare services organization also play a role? We agree to expand this section with more detailed analysis and add comparison with similar studies.

Answer to Question 4: We agree with this remark. Our research is similar to other researchers conducted worldwide. The study shows that the outcome of infection with COVID -19 depends on the age, sex and the number of comorbidities as well as on the type of hospitalization: emergent or planned.  From the clinical perspective, this will help clinicians in decision making (at present monoclonal antibodies are available for the high risk groups and it should be used as soon as possible), will guide them when choosing the proper treatment. It is now known that mortality is higher in men compared to women, in the older age groups, when the patients are hospitalized upon emergency up to the 6 th hour and when the number of comorbidities is high, especially cardiovascular disorders and diabetes. Moreover, the study illustrates that the access to high quality medical treatment and the organization of healthcare also plays a role in the disease outcome. We believe that the number and the character of comorbidities are important predictors of the infection outcome and will aid in development of reliable severity scores similar to those for pneumonia.

Sincerely,

All authors

Reviewer 2 Report

I have reviewed the manuscript authored by Dimova et. al titled " COVID-19 mortality and related comorbidities in hospitalized patients in Bulgaria. The paper is not well written and structured. In my opinion the paper has some shortcomings in regards to some data analyses and text, and I feel the dataset has not been utilized to its full extent. Furthermore I have made additional suggestions for more in-depth analyses of the data. Key critical points are:

1.     Methodology, which is a crucial component, is written very inadequately. Not even the inclusion and exclusion criteria are specified in detail.

2.     It is also not indicated how long the patients' clinical outcomes were monitored.

3.     There is ambiguous description of case definitions, i.e. hospitalized patients should be reprted into two groups ‘‘Nonsevere,’’ cases with mild and moderate clinical forms and ‘‘Severe,’’ the patients with severe and critical clinical forms.

4.     Laboratory results of the patients with COVID-19 at admission should also be provided as main or supplementary data.

5.     It is suggested to include the flowchart showing the strategy of participants’ enrollment, that would be much informative rather than depicting the same via text.

6.     A detailed “Supplementary Information” section might be useful and could help to keep the manuscript in a short format.

7.     The entire manuscript has to be revised in terms of language and grammar.

8.     Representation of the existing data in the manuscript is also very poor.

9.     Surprisingly, details of the Statistical analysis used, is also missing.

Given these shortcomings the manuscript requires major revisions.

Author Response

Dear Editors,

Thank you for the detailed reviews and the suggested amendments to out article.

We are grateful to reviewer 2 for his remarks. We shall take them all into consideration. In response to the questions, we would like to state:

Answer to Question 1 and 2. No, the data set is not utilized to its full extent. However, the age, sex and the number and character of comorbidities are among the first indicators, seen by the clinicians upon admission, even before the history, clinical features and before the laboratory and imaging test results are ready. This will help initially to stratify the patients and to some extent predict the outcome. Regarding the case definitions, the PCR for COVID - 19 became widely available in Bulgaria in the winter of 2019 and the antigen test - in mid 2020. The study covers the period from April 2020 to November, 2021. The inclusion criteria were the WHO case definitions of confirmed case of SARS-Co-2: either 1. a patient with positive NAAT test regardless of the clinical criteria or epidemiological criteria or 2. a patient meeting clinical criteria and /or epidemiological criteria (suspect case) with a positive professional use or self – test SARS-Co-2Antigen RDT.  There are no exclusion criteria. We have already specified this in the Material and Methods section of the revised manuscript.

Answer to Question 3. The article reports only hospitalized patients with COVID-19. St. George is the university hospital with specialized ICU for patients with COVID-19, Clinics of Infectious Diseases with approximately 150 beds and a specialized Pulmonology Ward. In Plovdiv, no other hospital has these facilities available. Therefore, mostly severe and critical cases were admitted – meaning pneumonia confirmed by either chest X rays or CTs, indicating extension to the alveoli. Also moderately ill patients at high risk were also admitted. We have specified this in the manuscript.

Answer to Question 4. We agree that the laboratory tests are important. However, the reported results correspond to the objective and the title of the manuscript. Laboratory data are important, but providing them in full detail is not possible. The laboratory works up of patients with COVID -19 depends on the patients’ condition. Would you please specify which laboratory tests should be included: the complete blood count and differential, the biochemical test with CRP, LDH, ferritin renal and liver parameters, or coagulation studies including fibrinogen and d- dimers? In critically ill patients, arterial ABG should also be done. The presence of comorbidities would also modify the laboratory test, performed – for instance glucose profile, cardiac enzymes, etc. For this reason, in our opinion, inclusion of laboratory tests upon admission in full is not possible.

Answer to Question 5. We agree with this remark. A flow chart will be more informative. We have already included it Material and methods in the revised version.

Answer to Question 6. We beg to differ. This would result in unnecessary repetition of data in the manuscript.

Answer to Question 7. The manuscript would be checked by a native English speaker.

Answer to Question 8 and 9. The statistical methods used are outlined in the Materials and Methods section. In the Results section, each obtained result is presented, followed by a specified statistical method, used.

Sincerely,

All authors

Round 2

Reviewer 2 Report

I have reviewed the manuscript again.  The work has, in my judgement, been extremely well corrected by the authors.

I suggest that this manuscript be considered for publication.

Thanks